# Contact Engineering Approach to Improve the Linearity of Multilevel Memristive Devices

**DOI:** 10.3390/mi12121567

**Published:** 2021-12-16

**Authors:** Natalia Andreeva, Dmitriy Mazing, Alexander Romanov, Marina Gerasimova, Dmitriy Chigirev, Victor Luchinin

**Affiliations:** Department of Micro- and Nanoelectronics, St. Petersburg Electrotechnical University ‘LETI’, Saint Petersburg 197376, Russia; dmazing@yandex.ru (D.M.); event-horizon@mail.ru (A.R.); gerasimova.m.i@mail.ru (M.G.); dachigirev@mail.ru (D.C.); cmid_leti@mail.ru (V.L.)

**Keywords:** multilevel memristor, metal oxide thin films, atomic layer deposition, contact engineering

## Abstract

Physical mechanisms underlying the multilevel resistive tuning over seven orders of magnitude in structures based on TiO_2_/Al_2_O_3_ bilayers, sandwiched between platinum electrodes, are responsible for the nonlinear dependence of the conductivity of intermediate resistance states on the writing voltage. To improve the linearity of the electric-field resistance tuning, we apply a contact engineering approach. For this purpose, platinum top electrodes were replaced with aluminum and copper ones to induce the oxygen-related electrochemical reactions at the interface with the Al_2_O_3_ switching layer of the structures. Based on experimental results, it was found that electrode material substitution provokes modification of the physical mechanism behind the resistive switching in TiO_2_/Al_2_O_3_ bilayers. In the case of aluminum electrodes, a memory window has been narrowed down to three orders of magnitude, while the linearity of resistance tuning was improved. For copper electrodes, a combination of effects related to metal ion diffusion with oxygen vacancies driven resistive switching was responsible for a rapid relaxation of intermediate resistance states in TiO_2_/Al_2_O_3_ bilayers.

## 1. Introduction

The fast growth of data that have to be processed by modern computing systems makes it necessary to search for effective solutions for increasing computing performance. A rapidly developing concept of memory-centric architectures, so-called in-memory and near-memory computing [1], promises to boost the performance of computing systems, overcoming memory bottlenecks of the von Neumann architectures. The goal of these computing design approaches is to minimize data movement as much as possible, performing the computations inside the memory (for in-memory computing) or bringing the memory close to logic chips as much as possible to maximize data bandwidth (for near-memory computing). The last approach requires the development of new processor-memory interfaces [2]. While for the first one, established memory technologies, such as SRAM, DRAM, ROM/RAM, or flash, could be used, or alternative memory types could be developed. For the established memory technologies, scaling remains the main issue for further computing transformation based on existing memory types. This circumstance gives rise to the development of the next-generation non-volatile memory technologies, such as ferroelectric field-effect transistor memory (FeFET), magnetoresistive random-access memory (MRAM), phase-change memory (PCV), and resistive random access memory (ReRAM). All of them are attractive combining the speed of SRAM and non-volatility of flash. Besides overcoming the von Neumann bottlenecks, memory-centric architectures substantially simplify the development of the modern non-traditional computation approach, or neuromorphic computing, which is based on mimicking the way that our brain uses for data processing [3]. In this approach, hardware-implemented neural networks perform computations on large amounts of data. It was demonstrated, that in-memory computing allows increasing application performance of machine learning algorithms in neuromorphic computing due to efficient implementation of fundamental operations of any typical neural network such as matrix-vector multiplication (MVM) [4,5]. For the moment, it is difficult to say which is the best memory type for in- and near-memory computing, but one of the most promising next-generation memory for neuromorphic computing is considered to be ReRAM [6,7]. Its integration into crossbar arrays offers highly-parallel and efficient hardware realization of MVM operations [8,9]. Among advantages of ReRAM devices are its extreme scaling, confirmed by fabricating devices with an electrode size in the range of 23 nm [10], low leakage, and potentially wide memory window (the experimentally demonstrated R_ON_/R_OFF_ ratio reaches 10^11^ [11]) with a possibility to obtain intermediate states between the R_ON_ and R_OFF_ resistance. The high OFF/ON ratio provides a better sensing capability for operating in circuits and enables using large-scale crossbar arrays in hardware architecture. The intermediate resistance states offer the basis for multilevel logic, which is important for achieving high-density storage [12]. The main shortcomings of ReRAM are coming from relatively high write energy (2 nJ) and latency (100 ns), and, for some devices, their poor endurance [13]. In addition, one of the basic requirements for the successful implementation of ReRAM crossbars for MVM in neuromorphic computing systems is device linearity. Unfortunately, despite all the advantages, current ReRAM devices are highly nonlinear, which makes write operation and peripheral circuitry implementation in computing systems expensive.

Recently, we developed an approach to design thin-film TiO_2_/Al_2_O_3_ bilayer structures [14,15], exhibiting electric-field analog tuning of the nonvolatile resistance state in the range of seven orders of magnitude. Despite a wide memory window and the existence of multiple nonvolatile resistance states, TiO_2_/Al_2_O_3_ bilayer structures demonstrate nonlinear behavior originating from the initial difference in the resistive properties of TiO_2_ and Al_2_O_3_ layers. resulting in the nonlinear characteristics of the analog tuning [16]. Moreover, the presence of hydroxyl groups in functional oxide layers of our structures could not be eliminated due to the atomic layer deposition (ALD) technique, employed in the device fabrication. In combination with platinum electrodes, the development of oxygen-related electrochemical reactions involving OH-groups occurs at the interface with electrode regions owing to the catalytic activity of the platinum. The impact of oxygen-related electrochemical reactions on the oxygen vacancy density in the Al_2_O_3_ layer should also unpredictably contribute to the nonlinear behavior of TiO_2_/Al_2_O_3_ bilayer structures [17].

We assume that contact engineering could help to improve the linearity of the analog tuning of the resistance of our bilayer devices. Organization of the controlled electro-oxidation reactions between the metal electrode and the active (switching) oxide layer (in our case, Al_2_O_3_) of the bilayer structure may be considered as an additional capacity for oxygen-related ions, which could enhance the linearity of analog tuning. For this purpose, we replace the material of the top platinum electrode of TiO_2_/Al_2_O_3_ bilayers with chemically active materials, such as copper and aluminum. The choice of electrode materials was guided by their activity, based on the value of standard electrode potential. Aluminum is a good reducing agent with the negative value of the standard reduction potential (−1.66 V) and has a strong affinity to oxygen, while copper acts as an oxidizing agent with the positive value of standard electrode potential (+0.34 V) and is reduced to metal in aqueous electrolytes. Moreover, copper has been widely used as an electrode material in the metal-ion-based conductive bridge resistive random access memory (CBRAM), serving as a source of ions for conductive filament formation. This paper reports the results of the investigation of the effect of the electrode material on the resistive properties of TiO_2_/Al_2_O_3_ bilayer structures aiming to enhance the linearity of TiO_2_/Al_2_O_3_ bilayers while maintaining the wide memory window of the structures and keeping the possibility of analog tuning of the nonvolatile resistance state of the devices. 

## 2. Materials and Methods

A 50 nm thick bottom platinum electrodes (Pt-BE) were deposited on a p-type Si/SiO_2_ substrate with a 10 nm thick titanium adhesive layer by DC magnetron sputtering at T = 150 °C. TiO_2_ (30 nm)/Al_2_O_3_ (5 nm) bilayers were grown via ALD on the mentioned substrate with TFS 200 system (Beneq, Espoo, Finland) at 150 °C using trimethylaluminum (Al(CH_3_)_3_) and titanium isopropoxide (Ti[OCH(CH_3_)_2_]_4_) as precursors and water vapor as an oxidizing agent. The thickness of the layers in Si/SiO_2_/Ti/Pt/TiO_2_/Al_2_O_3_ structures was controlled by scanning electron microscopy applied to a cross-section formed by focused ion beam (FEI, Helios NanoLab, Hillsboro, OR, USA). The surface topography of metal oxide layers was studied using atomic force microscopy (AFM) (Dimension 3100, Veeco, New York, NY, USA). In all fabricated metal-oxide bilayers, Al_2_O_3_ layers are amorphous. Post-deposition annealing was done at 200 °C for 30 s under ambient atmosphere. 50 nm thick Pt top electrodes (Pt-TE) and 100 nm thick Cu/Ni top electrodes (Cu-TE) were deposited by magnetron sputtering at T = 150 °C. 100 nm thick Al top electrodes (Al-TE) were deposited using electron beam evaporation at T = 150 °C. The top electrodes were patterned using a stencil mask, whose apertures were with the diameter of 350 μm for Pt-TE and Al-TE and 50 μm for Cu-TE (Figure 1).

To verify the resistive properties and the role of every single metal-oxide layer in TiO_2_/Al_2_O_3_ bilayers, the Si/SiO_2_/Ti/Pt(50 nm)/TiO_2_(30 nm)/Pt(50 nm) and Si/SiO_2_/Ti/Pt(50 nm)/Al_2_O_3_(10 nm)/Pt(50 nm) structures were fabricated by the above-described method. To investigate resistive switching in the fabricated bilayer structures, *I-V* curves were measured using a Keithley 4200-SCS (Keithley Instruments Inc., Solon, OH, USA) semiconductor characterization system at ambient conditions using a two-probe configuration. For *I-V* measurements, tungsten needles were put in contact with the top and bottom electrodes of the structures. The operating voltage was applied to the top electrodes whereas the bottom electrodes were grounded. The resistance of structures was measured by using low (0.1 V) dc voltage.

## 3. Results and Discussions

In Pt-BE/TiO_2_/Al_2_O_3_/Pt-TE bilayer structures, 5 nm thick Al_2_O_3_ layer plays a role of an active (or switching) layer, while 30 nm thick titanium oxide layer acts as a reservoir of oxygen vacancies (Figure 2). 

Under bias voltage application, oxygen vacancies drift in the Al_2_O_3_ layer, causing the reversible modification [18] of its properties and setting the resistance state of the structure. Relatively to a given resistance state determined by the value of bias voltage, a bipolar resistive switching takes place. An appearance of bipolar resistive switching is associated with electron-like processes related to the dominant transport mechanism in materials with high trap concentrations and accompanied by current pinching arising from the fluctuation instability in a direction perpendicular to the current [19]. The resistance of the TiO_2_/Al_2_O_3_ bilayer structures in the pristine state is 0.8 × 10^12^ Ω and is provided by the Al_2_O_3_ layer (the resistivity of Al_2_O_3_ is 10^13^–10^15^ Ω·cm as opposed to 10^4^–10^7^ Ω·cm for TiO_2_). In an analog tuning, the resistive properties of aluminum oxide are modified in a broad range (seven orders of magnitude) due to the variation of oxygen vacancy density [20]. The summary of resistive switching properties of TiO_2_/Al_2_O_3_ bilayer structures in comparison with the experimentally reported realization of multilevel memristors is presented in Table 1.

An appearance of the analog tuning regime in TiO_2_/Al_2_O_3_ bilayers is attributed to the properties of the TiO_2_ layer. In this case, it could be expected, that single-layer structures with a TiO_2_ switching layer (Figure 3a) should also be able to demonstrate an analog tuning behavior, coupled with the characteristic counterclockwise bipolar switching behavior (Figure 3b) and associated with the gradual decrease of the oxygen vacancy density [29]. Considering, that a bipolar resistive switching in Pt/TiO_2_(30 nm)/Pt structures are attributed to the formation of the conductive filamentary (CF) area, the gradual resistance tuning might be related to the variation of the oxygen vacancy concentration in the CF, formed during the SET process (switching from high resistance state (HRS) to low resistance state (LRS)). Indeed, the second nonvolatile resistive switching process with the switching direction of opposite polarity (i.e., clockwise) could be activated from the LRS of Pt/TiO_2_/Pt structures (Figure 3c) by applying a voltage to the Pt-TE less negative than the RESET value. The structure undergoes the transition into the intermediate resistance state, which could be tuned by applying a positive voltage to the Pt-TE less positive than the SET value. This additional clockwise bipolar resistive switching has a significantly lower value of R_OFF_/R_ON_ ratio (Figure 3d) and could be attributed to the oxide’s defects redistribution in the TiO_2_ layer, mainly in the CF/TE interface area, due to electrochemical reactions involving OH-groups [30,31]. This way, positively charged Pt-TE attracts negatively charged ions, such as OH^−^, O_2_^2−^, triggering the electrochemical reaction (the value of standard electrode potential [32] is given in parentheses):$${4\mathrm{OH}}^{-}{-\;4e}^{-}\;\to \;2H_{2}O+2O\;(-0.401\;V)$$

The developed oxygen reduction reaction induces the oxidation of Ti in a TiO_2_ switching layer, and reduces the concentration of oxygen vacancies related to Ti^3+^ in the next possible way [33]:$${\mathrm{Ti}}^{3+}{-\;e}^{-}\to {\;\mathrm{Ti}}^{4+}\;(+0.092\;V)$$

This leads to the increase of the stoichiometry of TiO_2–x_ in the CF area at the interface with the Pt-TE and results in increasing the structure resistance, i.e. switching to the HRS. The clockwise switching to the LRS takes place under the reverse polarity of the voltage bias and is accompanied by the electrochemical reactions with Ti^4+^ reduction to Ti^3+^ state. 

When the conductivity of the HRS for a clockwise resistive switching reaches the value of HRS for the counterclockwise bipolar switching, the unipolar switching to the LRS of the counterclockwise switching mode occurs. Both the clockwise and the counterclockwise resistive switching modes do not obstruct each other. Moreover, the clockwise resistive switching leads to an appearance of intermediate resistance states in the range between the LRS and HRS of the counterclockwise bipolar switching.

It should be noted, that despite the existence of intermediate states in single-layer Pt/TiO_2_/Pt structures, the highest possible memory window of these systems is limited by the R_OFF_/R_ON_ ratio of the counterclockwise bipolar resistive switching and does not exceed two-three orders of magnitude.

Adding the Al_2_O_3_ layer to the TiO_2_ layer allows extending the range of the nonvolatile tuning of the structural resistance to seven orders of magnitude. Thus, in Pt-BE/TiO_2_/Al_2_O_3_(10 nm)/Pt-TE bilayer structures, aluminum oxide plays the role of an active (switching) layer [15]. This statement is supported by experimental observation of the resistive switching in single-layer Pt/Al_2_O_3_/Pt structures with an R_OFF_/R_ON_ ratio of about seven orders of magnitude (Figure 4). More details of the resistive switching in single-layer Pt/Al_2_O_3_/Pt structures can be found elsewhere [17].

The conductivity of Pt-BE/TiO_2_/Al_2_O_3_/Pt-TE bilayer structures is related to trap-assisted space-charge-limited currents (SCLC) in high-resistivity materials with low carrier mobility and long dielectric relaxation time. Traps or localized states within the Al_2_O_3_ bandgap are formed by the oxygen vacancies. The SCLC transport mechanism is manifested in power dependence of current on voltage with a changeable degree of power [15]. Both bipolar and multilevel resistive switching are clockwise for these structures. The first bipolar resistive switching, associated with the resistance state with the lowest conductivity (HRS-1 in Figure 2), is accompanied by the formation of the filamentary conductive area in the Al_2_O_3_ layer. SET process of this bipolar resistive switching happens at high injection levels, at which the traps are filled, reaching the trap-filled limit (TFL) [19]. Thus, the experimentally observed *I-V* curves for HRS are linearized in a double logarithmic scale with several characteristic parts, corresponding to linear, square, and power dependences (with n ≥ 3). While for LRS, the transition to the trap-free-square-law region takes place and experimental *I-V* curves exhibit only linear and square dependences. The limiting condition for the development of the conductive filament in the Al_2_O_3_ layer is a transition to the TFL regime, i.e., the trap filling process supplies positive feedback for the filament development [19]. 

An appearance of the next conductive state in the Pt-BE/TiO_2_/Al_2_O_3_/Pt-TE bilayer structures is related to an increase of the oxygen vacancy concentration in the filamentary conductive area of the Al_2_O_3_ due to the oxygen vacancies drift in from the TiO_2_ layer. In general terms, discretization of resistance states is defined by the statistically significant difference in the value of the resistivity (ρ). For the hopping conductivity, the resistivity could be estimated based on percolation theory [34] and is given by: $\rho =\rho_{0}\cdot \exp\left[\frac{1.73}{N^{\frac{1}{3}}\cdot a_{B}}\right]$ (where $a_{B}$ is Bohr radius and N is the oxygen vacancy concentration). Thus, the resistivity of the bilayer structure nonlinearly depends on the oxygen vacancy density in the CF area in the Al_2_O_3_ layer. The voltage drop across the Al_2_O_3_ layer in a bilayer structure is determined by the ratio of the resistance of TiO_2_ and Al_2_O_3_ layers (the resistivity of Al_2_O_3_ is 10^13^–10^15^ Ω cm as opposed to 10^4^–10^7^ Ω cm for TiO_2_), while the resistance of the Al_2_O_3_ layer is driven by the field-controlled oxygen vacancy drift and nonlinearly depends on the oxygen vacancy concentration in the Al_2_O_3_ layer. The combination of these factors results in the nonlinear characteristics of the tuning of the resistance state in Pt-BE/TiO_2_/Al_2_O_3_/Pt-TE bilayer structures.

In Pt-BE/TiO_2_/Al_2_O_3_/Al-TE bilayer structures, an impact from electrochemical reactions involving OH-groups at the interface with the top electrode is expected and can be defined in general as
$${\mathrm{Al}+4\mathrm{OH}}^{-}{-\;3e}^{-}\to {\mathrm{AlO}}_{2\;}^{-}{+2H}_{2}O\;(+2.35\;V)\;\mathrm{or}\;{\mathrm{Al}+3\mathrm{OH}}^{-}{\;-\;3e}^{-}\to {\mathrm{Al}(\mathrm{OH})}_{3}\;(+2.31\;V)$$

Indeed, the experimental *I-V* curves provide evidence of memristive behavior starting from the values of the applied voltage, which are approximately equal to the standard electrode potentials of reactions, mentioned above. The resistance state of the structures is gradually decreasing with increasing the value of the applied voltage above the standard electrode potentials (Figure 5a). Analog reversible tuning of the resistance is observed within the range of more than three orders of magnitude until the current through the bilayer structure exceeds the level of 10 mA. At this current, the switching to the LRS, associated with the characteristic counterclockwise bipolar switching behavior of the TiO_2_ layer, happens (Figure 5a). 

Relatively to the given resistance state, tuned by applying a positive voltage to the Al-TE and determined by its value (2.3 to 4 V), a bipolar counterclockwise resistive switching takes place, accompanied by the next electrochemical reactions
$${\mathrm{Al}(\mathrm{OH})}_{3}{+3H}^{+}{+3e}^{-}\to {\mathrm{Al}+3H}_{2}O\;(-1.47\;V)\;\mathrm{or}\;{\mathrm{AlO}}_{2\;}^{-}{+4H}^{+}{+3e}^{-}\to {\mathrm{Al}+2H}_{2}O\;(-1.26\;V)$$

The R_OFF_/R_ON_ ratio for this bipolar resistive switching varies with the resistance state of the structure from 1.5 to 5.5 (Figure 5b). The highest value of the R_OFF_/R_ON_ ratio corresponds to the intermediate range of the analog tuning, while at the edges of the tuning range the R_OFF_/R_ON_ ratio decreases to 1.5. *I-V* characteristics for both HRS and LRS are nonlinear, asymmetrical (regarding the polarity of applied bias), and could be linearized in a double logarithmic scale with several characteristic parts ($I\sim U^{n}$), corresponding to linear and power dependences with n = 3. Compared to Pt-BE/TiO_2_/Al_2_O_3_/Pt-TE bilayer structures, for which the degree of power dependence of the corresponding part of *I-V* curves for HRS differs from those for LRS, Pt-BE/TiO_2_/Al_2_O_3_/Al-TE bilayer structures exhibit the same degree of power dependence for both HRS and LRS (Figure 5c). Previous experimental evidence [15,19] suggests the dominant role of electronic processes in the bipolar resistive switching in bilayer structures with platinum electrodes. The transition from the HRS to the LRS is associated with reaching the trap-filling limit (TFL) in the local conductive filamentary area at a high injection level. At reaching TFL, the transition to the trap-free-square law takes place and the *I-V* curves for LRS follow the square law, while for HRS the characteristic part for power dependence has a degree of power n ≥ 3. The same degree of power dependence for both HRS and LRS in Pt-BE/TiO_2_/Al_2_O_3_/Al-TE bilayer structures indicates changes in the mechanism of the bipolar resistive switching from electronic one to electrochemically driven. Reducing the R_OFF_/R_ON_ ratio from 1–2 orders of magnitude for structures with Pt-TE to less than 10 for Al-TE provides further evidence in support of this assumption. However, it should be noted, that dominance of electrochemically driven processes in the bipolar resistive switching of bilayer structures with Al-TE brings about linearity in the dependence of the resistance state on the voltage at the analog tuning of the resistance.

Copper is widely used as an electrode material in CBRAM devices due to its high metal diffusivity with low activation energy. At the same time, Al_2_O_3_ has advantages when used in conjunction with Cu electrodes, allowing to control the Cu migration and enhance the device endurance. The high energy gap of Al_2_O_3_ ensures the maintenance of the high R_OFF_/R_ON_ ratio, whereas its high thermal conductivity (15 W/mK) facilitates the formation of Cu ion conductive filament [35]. In addition, oxidation of Cu electrode at the Al_2_O_3_/Cu-TE interface provides a higher migration rate of Cu^2+^ ions due to the lower Cu-O bond energy compared with that of Cu-Cu metallic bond (1.5 eV vs. 2.0 eV [36]). 

For TiO_2_/Al_2_O_3_ bilayer structures, contributions from the competition of oxygen vacancies and Cu ions to the formation of the conductive filamentary area in the Al_2_O_3_ layer are expected to take place. Indeed, if the thickness of the Al_2_O_3_ layer does not exceed 5 nm, the gradual resistance tuning is experimentally observed at negative bias applied to the Cu-TE and could be associated with the formation of oxygen vacancy CF. For bilayer structures with larger thicknesses, tuning of the resistance takes place at the positive polarity of applied bias and is related to the formation of Cu ion CF (Figure 6a). 

The development of pinched *I-V* hysteresis occurs starting with sufficiently high voltage bias (5 V), indicating the impact of thermal or thermochemical processes associated with ion diffusion. 

Interestingly, at bipolar resistive switching in Pt-BE/TiO_2_/Al_2_O_3_/Cu-TE bilayer structures, relative to the given resistance state, an appearance of the characteristic part of *I-V* curves with negative differential resistance is experimentally observed on the LRS branch of *I-V* curves corresponding to the polarity of reset process (Figure 6b). We assume, that the region of the *I-V* curve with a negative slope (for 5 nm-thick Al_2_O_3_ layer) reflects the situation when the oxygen vacancy CF dissolves faster than Cu ion CF forms.

The switching to the LRS, associated with the characteristic counterclockwise bipolar switching behavior of the TiO_2_ layer, happens when the level of the current through the bilayer structure exceeds 1 mA and only at the positive polarity of applied bias. It should be noted, that the range of applied voltages for the resistance tuning of this type of structure is shifted toward higher values (from 5 to 7 V), in comparison with Pt-BE/TiO_2_/Al_2_O_3_/Al-TE bilayer structures. At the same time, the R_OFF_/R_ON_ ratio for bipolar resistive switching does not significantly vary with the resistance of the structure and is approximately equal to 5 (Figure 6c).

Despite the improved linearity of the resistance tuning, the major shortcoming of Pt-BE/TiO_2_/Al_2_O_3_/Cu-TE bilayer structures is a rapid degradation of intermediate resistance states with time (relaxation effect) (Figure 7), attributed to an impact from the electrochemical reactions at the Al_2_O_3_/Cu-TE interface: $$2\mathrm{Cu}+2{\mathrm{OH}}^{-}\;-\;2e^{-}\to {\mathrm{Cu}}_{2}O\;+\;H_{2}O\;\left(+0.36\;V\right)\;\mathrm{Cu}\;+\;2{\mathrm{OH}}^{-}\;-\;2e^{-}\to {\mathrm{Cu}(\mathrm{OH})}_{2}\;\left(+0.22\;V\right)\;{\mathrm{Cu}}_{2}O\;+\;2{\mathrm{OH}}^{-}\;+\;H_{2}O\;-\;{2e}^{-}\to 2\mathrm{Cu}{\left(\mathrm{OH}\right)}_{2}\;\left(+0.08\;V\right)$$

The values of standard electrode potential for these reactions suggest that in the presence of OH-groups in the Al_2_O_3_ layer copper oxidation/reduction processes strongly affect the original structure of conductive filament. 

In summary, according to the experimental results, the material of the top electrode significantly influences the mechanism and operating parameters of nonvolatile resistive switching in TiO_2_/Al_2_O_3_-based bilayer structures. In the case of Al-TE, the prevalence of oxygen-related electrochemical reactions suppresses the development of the electronic mechanism of resistive switching, previously observed for Pt-BE/TiO_2_/Al_2_O_3_/Pt-TE structures. Switching to the electrochemical mechanism of resistive tuning results in the narrowing of a memory window down from seven to three orders of magnitude, while improving the linearity of the resistance tuning at the same range of operation voltages. The specificity of Cu-TE leads to an appearance of mixed effects, induced by Cu ion migration in the active Al_2_O_3_ layer of the structure on the one side, and oxygen vacancy-related formation of CF in the Al_2_O_3_ layer, on the other side. The combination of these competitive effects strongly deteriorates the performance of the TiO_2_/Al_2_O_3_ bilayer structures, shifting the operating voltages to higher values and causing the rapid relaxation of the intermediate resistance states in Pt-BE/TiO_2_/Al_2_O_3_/Cu-TE structures.

Comparison of the device performance between TiO_2_/Al_2_O_3_ bilayer structures with Al-TE and Cu-TE and those from recent reports (Table 1) shows that despite narrowing of the memory window when using Al as a top electrode material, the R_OFF_/R_ON_ ratio for those devices remains the highest among experimentally demonstrated multilevel memristors with analog tuning between the nonvolatile resistance states. According to the same Table 1, an attempt to combine the Cu ion migration together with oxygen-related electrochemical reactions at the top electrode interface in Pt-BE/TiO_2_/Al_2_O_3_/Cu-TE structures brings no benefits to the device performance. 

## 4. Conclusions

Concluding, electrochemical reactions involving OH-groups could be experimentally observed in ALD deposited aluminum and titanium oxide layers in both single-layer and bilayer structures with resistive switching effects. In single-layer Pt/TiO_2_/Pt structures oxygen-related electrochemical reactions induce an analog tuning from LRS (50 Ω) to HRS (50 kΩ). Single-layer Pt/Al_2_O_3_/Pt structures show the abrupt transition from HRS (~10^12^ Ω) to LRS (~10^4^–10^3^ Ω) without controllable switching to intermediate resistance states. The resistance state of the combination of two oxide layers (TiO_2_ and Al_2_O_3_) in a bilayer structure, sandwiched between platinum electrodes, can be electrically tuned in an analog manner over seven orders of magnitude (from ~10^12^ Ω up to ~10^5^ Ω). The resistance of intermediate states in bilayer structures is determined by the concentration of oxygen vacancies in the active layer of Al_2_O_3_. Unfortunately, the initial difference in the resistive properties of TiO_2_ and Al_2_O_3_ layers causes the nonlinear dependence of the state conductivity on the switching voltage in the analog regime. The contribution of the electrochemical reactions at the Al_2_O_3_/TE interface, induced by top electrode material substitution, may improve the linearity of analog nonvolatile tuning, but gives rise to the transformation of the resistive switching mechanism of bilayer structures. This, in turn, results in a separate observation of resistive switching effects, attributed to either Al_2_O_3_ or TiO_2_ layers of bilayer structures. Consequently, the range of the nonvolatile analog tuning of the resistance has narrowed down to three orders of magnitude (from ~10^6^ Ω to ~10^3^ Ω) in the case of Al-TE, together with the decreasing of the R_OFF_/R_ON_ ratio of the bipolar resistive switching (relatively to the given resistance state) from one-two orders of magnitude to 5. A combination of effects related to metal ion diffusion with oxygen vacancies driven resistive switching in TiO_2_/Al_2_O_3_ bilayers with Cu-TE, in addition to narrowing of the range of analog tuning (from ~10^8^ Ω to ~10^7^–10^6^ Ω), provokes a rapid relaxation of intermediate resistance states, shifting switching voltages toward higher values (from 2–4 V for bilayers with Pt-TE and Al-TE to 6–7 V). 

## Figures and Tables

**Figure 1 micromachines-12-01567-f001:**
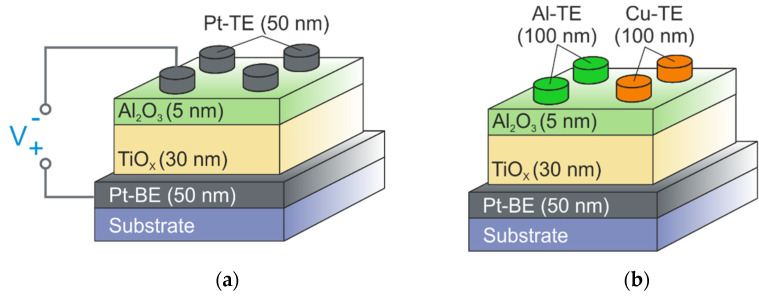
Schematic illustration of bilayer structures. (**a**) Si/SiO_2_/Ti/Pt-BE/TiO_2_/Al_2_O_3_ structures with platinum top electrodes; (**b**) Si/SiO_2_/Ti/Pt-BE/TiO_2_/Al_2_O_3_ structures with aluminum and copper top electrodes.

**Figure 2 micromachines-12-01567-f002:**
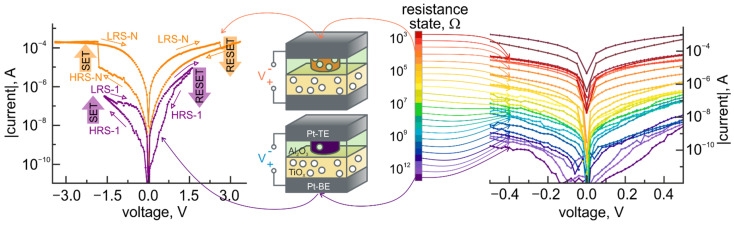
Experimental *I-V* characteristics of Pt/TiO_2_/Al_2_O_3_/Pt structures with a combination of an electric-field analog tuning and a bipolar resistive switching and the schematic illustration of underlying physical mechanisms.

**Figure 3 micromachines-12-01567-f003:**
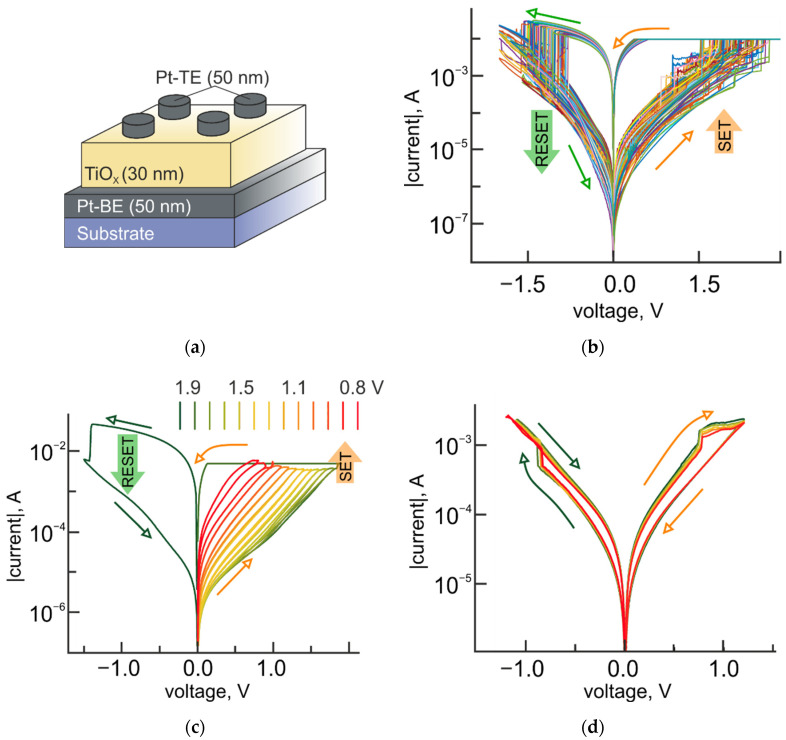
Resistive switching modes in Pt/TiO_2_/Pt structures. (**a**) Schematic illustration of Pt/TiO_2_/Pt structure; (**b**) Experimental *I-V* curves for the counterclockwise filamentary-type bipolar resistive switching; (**c**) Experimental observation of the coexistence of the counterclockwise and the clockwise bipolar resistive switching modes; (**d**) Experimental *I-V* curves for the clockwise bipolar resistive switching for 5 consequent switching cycles.

**Figure 4 micromachines-12-01567-f004:**
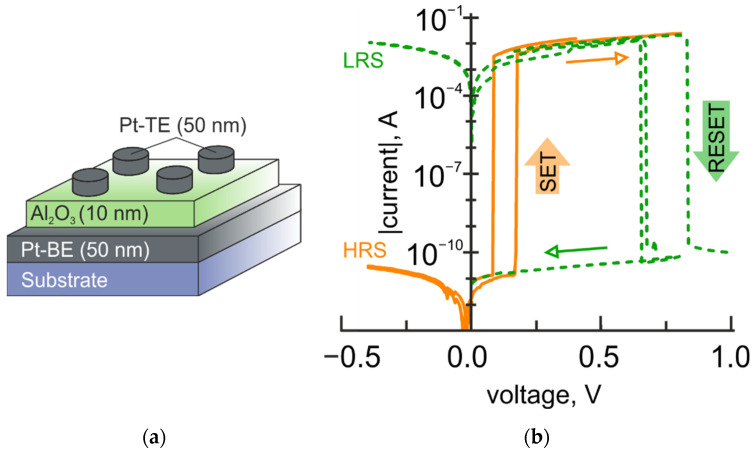
Resistive switching in Pt/Al_2_O_3_/Pt structures. (**a**) Schematic illustration of Pt/Al_2_O_3_/Pt structures; (**b**) Experimental *I-V* curves for the resistive switching in Pt/Al_2_O_3_/Pt structures with a memory window of seven orders of magnitude.

**Figure 5 micromachines-12-01567-f005:**
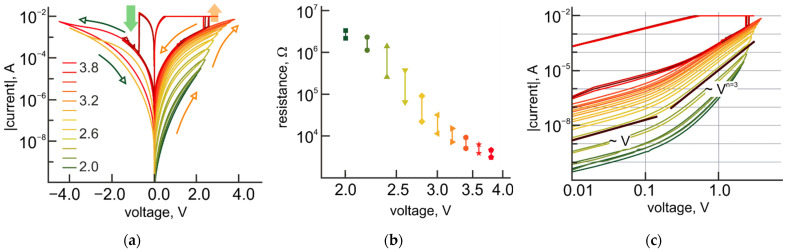
Resistive switching modes in Pt-BE/TiO_2_/Al_2_O_3_/Al-TE bilayer structures. (**a**) Experimental *I-V* curves for gradual resistance tuning as a function of voltage combined with a bipolar resistive switching in Pt-BE/TiO_2_/Al_2_O_3_/Al-TE structures; (**b**) Dependence of R_OFF_/R_ON_ ratio on the resistance state of Pt-BE/TiO_2_/Al_2_O_3_/Al-TE structures; (**c**) *I-V* curves from (**a**) in a double logarithmic scale.

**Figure 6 micromachines-12-01567-f006:**
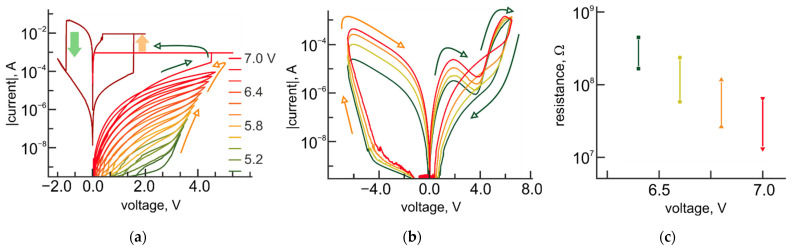
Resistive switching modes in Pt-BE/TiO_2_/Al_2_O_3_/Cu-TE bilayer structures. (**a**) Experimental *I-V* curves for a gradual resistance tuning as a function of voltage resistive switching in Pt-BE/TiO_2_/Al_2_O_3_/Cu-TE for structures with 7-nm thick Al_2_O_3_ layer; (**b**) Experimental *I-V* curves for the bipolar resistance switching in Pt-BE/TiO_2_/Al_2_O_3_/Cu-TE for structures with 5 nm-thick Al_2_O_3_ layer; (**c**) Dependence of R_OFF_/R_ON_ ratio on the resistance state of Pt-BE/TiO_2_/Al_2_O_3_/Cu-TE structures.

**Figure 7 micromachines-12-01567-f007:**
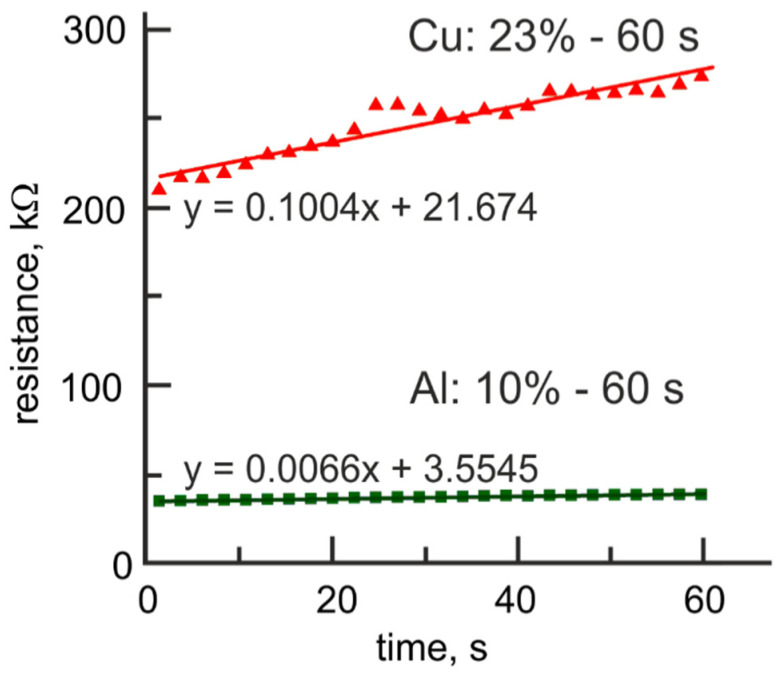
Time evolution of intermediate resistance state for Pt-BE/TiO_2_/Al_2_O_3_ bilayer structures with aluminum and copper top electrodes.

**Table 1 micromachines-12-01567-t001:** Comparison of the developed design of TiO_2_/Al_2_O_3_ bilayer structures with the performance of experimentally demonstrated multilevel memristors.

Memristor Geometry	Number of Resistance States	R_OFF_/R_ON_ or Conductance Range	Multilevel Control	SET Process	RESET Process	References
Pt/TiO_2_/AlxOy/Pt Pt/TiO_2_/TaxOy/Pt Pt/TiO_2_/WOx/Pt Pt/TiO_2_/HfOx/Pt Pt/TiO_2_/ZnOx/Pt	47 ≈36 ≈32 ≈22 ≈16	≈2.3 ≈1.3 ≈2.8 ≈1.4 2	identical pulses	100 ns voltage pulses at 2 V	100 ns voltage pulses at −2 V	[21]
Pt/Al_2_O_3_/TiO_2_−x/Ti/Pt	analog tuning	12–142 μS	identical pulses	500 μs voltage pulsesat 1.3 V	500 μs voltage pulsesat −1.3 V	[22,23]
W/WOx/Pd/Au	analog tuning	<5 μS	identical pulses	100 μs voltage pulses at 1.4 V	100 μs voltage pulsesat −1.3 V	[24,25]
Ir/TiO_x_/TiN	4	10^4^	voltage sweep	1 μs voltage pulses with a height of −2.8 V	1 μs voltage pulses with a height of 2 V, 2.4 V, 2.8 V	[26]
Pt/TaO_x_/TiN	4	3.2	variation of switching current	increase of current compliance level at 50, 100, and 200 μA	decrease of current compliance level at 200, 100, and 50 μA	[27]
Pt/W/TaO_x_/Pt	6	≈10^3^	voltage sweep	DC operation (current-voltage sweep) in the range of −1.50 V to −2.25 V	200 ns voltage pulses at 1.5 V	[28]
Pt/TiO_2_/Al_2_O_3_/Pt	analog tuning	≈10^7^	DC operation	DC operation (current-voltage sweep) in the range of −1.9 V to −4.0 V	DC operation (current-voltage sweep) in the range of 4.0 V to −1.9 V	[14,15]
Pt/TiO_2_/Al_2_O_3_/Al	analog tuning	≈10^3^	DC operation	DC operation (current-voltage sweep) in the range of 2.0 V to 4.0 V	DC operation (current-voltage sweep) in the range of −2.0 V to −4.0 V	this report
Pt/TiO_2_/Al_2_O_3_/Cu	analog tuning	≈4	DC operation	DC operation (current-voltage sweep) in the range of 6.4 V to 7.0 V	DC operation (current-voltage sweep) in the range of −6.4 V to −7.0 V	this report

## Data Availability

The data that support the findings of this study are available from the corresponding author, upon reasonable request.

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
