# Peer review of "Contact Engineering Approach to Improve the Linearity of Multilevel Memristive Devices"

_micromachines, 2021, doi:10.3390/mi12121567_

Round 1

Reviewer 1 Report

The article entitled, “Contact engineering approach to improve the linearity of multilevel memristive devices” discusses a contact engineering approach to improve the linearity of the electric-field resistance tuning by replacing Pt top electrodes with Al and Cu of TiO2/Al2O3 bilayer structure. The manuscript can be considered for publication, but it needs some revisions that are provided below. My decision is a major revision at this stage.

  • Correct the following language related mistakes from manuscript
    1. Change the word “rapidly” with “rapid” in line 30
    2. Avoid the expressions such as “so called” in line 55
    3. Replace the word “sense” with “sensing” in line 69
    4. Remove the word “of” from “of all the advantages” in line 76
    5. Remove the word “oxide” from line 81-82
    6. Use the proper symbol of multiplication “×” instead of “*” in line 94
    7. Replace the word “electrodes” with “electrode” in line
    8. Presentation of chemical reactions is not proper in the entire manuscript
    9. The whole manuscript should be revised thoroughly to remove language related mistakes as there are many more left that are not possible to mention here.
  • Authors are advised to improve the introduction section by following the given guidelines:
    1. Reduce the first paragraph of introduction section. It seems to be dragged.
    2. The third and fourth paragraphs (line 79-111) of introduction should also be moved to conduction mechanism section. This discussion is not appropriate to add in introduction.
    3. Merge the last two paragraphs of introduction section (line 112-130)
    4. It is not possible to give complete insight of the background knowledge of memristive devices in the introduction section of this manuscript therefore, authors are advised to add the references of following comprehensive review papers on memristive devices that have been published recently. This will prove to be useful for the wide readership of micromachines.
      1. Rehman, M.M., Rehman, H.M.M.U., Gul, J.Z., Kim, W.Y., Karimov, K.S. and Ahmed, N., 2020. Decade of 2D-materials-based RRAM devices: a review. Science and technology of advanced materials, 21(1), pp.147-186.
      2. Rehman, M.M., ur Rehman, H.M.M., Kim, W.Y., Sherazi, S.S.H., Rao, M.W., Khan, M. and Muhammad, Z., 2021. Biomaterial-Based Nonvolatile Resistive Memory Devices toward Ecofriendliness and Biocompatibility. ACS Applied Electronic Materials, 3(7), pp.2832-2861.
  • Merge all the paragraphs of materials and methods section into a single paragraph. Formatting of whole manuscript should be revised.
  • The authors need to tell the readers about the thickness of the switching layer in the structure, the size of the memory cell, and the size of the substrate used. These are key parameters of a resistive RAM.
  • It is better to give some details of the electrical characterization set up along with the type of equipment used, ambient condition, etc.
  • The conduction mechanism of this device must be explained through energy band diagram.
  • What does each color represent in figure 1? A proper legend should be added in this figure.
  • How many switching cycles are presented in figure 3(b)?
  • Authors should add some numbers from the obtained data in the conclusion section instead of only writing general statements

Author Response

Dear Referee,

We highly appreciate your comments and time you spent for reviewing our manuscript. We have taken into account all the critical comments, replied to them, and revised the manuscript accordingly.

Below, we give a detailed, point-by-point reply to all comments. Corresponding changes made in the revised draft of the manuscript are highlighted in red.

  • Correct the following language related mistakes from manuscript
    1. Change the word “rapidly” with “rapid” in line 30
    2. Avoid the expressions such as “so called” in line 55
    3. Replace the word “sense” with “sensing” in line 69
    4. Remove the word “of” from “of all the advantages” in line 76
    5. Remove the word “oxide” from line 81-82
    6. Use the proper symbol of multiplication “×” instead of “*” in line 94
    7. Replace the word “electrodes” with “electrode” in line
    8. Presentation of chemical reactions is not proper in the entire manuscript

The whole manuscript should be revised thoroughly to remove language related mistakes as there are many more left that are not possible to mention here

English proofreading of the text was carried out according to the reviewer’s comments.

The typo in line 94 was corrected as the reviewer suggested.

The presentation of chemical reactions was corrected in the entire manuscript.

  • Authors are advised to improve the introduction section by following the given guidelines:
    1. Reduce the first paragraph of introduction section. It seems to be dragged.

We agree with this comment and have reduced the first paragraph of the introduction. The first and second paragraphs were merged.

    1. The third and fourth paragraphs (line 79-111) of introduction should also be moved to conduction mechanism section. This discussion is not appropriate to add in introduction.

We have rearranged the description of TiO2/Al2O3 bilayer structures in the Introduction and moved the description of transport mechanisms in in the Results and Discussions section.

The aim of our study is to understand the influence of the material of the top electrode on the resistive effects observed in TiO2/Al2O3 bilayer structures with multilevel resistive switching. For this purpose, the short description of the structure’s design was left in the Introduction.     

    1. Merge the last two paragraphs of introduction section (line 112-130)

The last two paragraphs of the Introduction were merged

    1. It is not possible to give complete insight of the background knowledge of memristive devices in the introduction section of this manuscript therefore, authors are advised to add the references of following comprehensive review papers on memristive devices that have been published recently. This will prove to be useful for the wide readership of micromachines.
      1. Rehman, M.M., Rehman, H.M.M.U., Gul, J.Z., Kim, W.Y., Karimov, K.S. and Ahmed, N., 2020. Decade of 2D-materials-based RRAM devices: a review. Science and technology of advanced materials, 21(1), pp.147-186.
      2. Rehman, M.M., ur Rehman, H.M.M., Kim, W.Y., Sherazi, S.S.H., Rao, M.W., Khan, M. and Muhammad, Z., 2021. Biomaterial-Based Nonvolatile Resistive Memory Devices toward Ecofriendliness and Biocompatibility. ACS Applied Electronic Materials, 3(7), pp.2832-2861.

We added suggested references in the Introduction according to the reviewer recommendation.

Merge all the paragraphs of materials and methods section into a single paragraph. Formatting of whole manuscript should be revised.

We revised the formatting of the whole manuscript.

In the Materials and Methods section, the first and the second paragraphs were merged, as they are devoted to the sample’s growth. The last two paragraphs are related to the sample’s characterization and were merged together.

  • The authors need to tell the readers about the thickness of the switching layer in the structure, the size of the memory cell, and the size of the substrate used. These are key parameters of a resistive RAM.

Thicknesses of the switching layers of the structures are described in the Materials and Methods section in the first paragraph and in order to facilitate the article perception are represented in Figure 1 (a,b), Figure 3(a) and Figure 4(a) now.

  • It is better to give some details of the electrical characterization set up along with the type of equipment used, ambient condition, etc.

Thank you very much for this comment. We provide the details of the electrical characterization with Keithley 4200-SCS semiconductor characterization system were added in the Materials and Methods section.

  • The conduction mechanism of this device must be explained through energy band diagram.

We agree with the Referee that the electron transport mechanism of the TiO2/Al2O3 bilayer structures must be explained through the energy band diagram. Such explanation had already been provided in our previous work [Energy band diagram is presented in Figure 7b in reference 14: Alekseeva, L.; Nabatame, T.; Chikyow, T. et al. Resistive switching characteristics in memristors with Al2O3/TiO2 and TiO2/Al2O3 bilayer. Jpn. J. Appl. Phys. 2016, 55, 08PB02]

The aim of this study was to introduce the electro-oxidation reactions between the metal electrode and the switching oxide layer of the TiO2/Al2O3 bilayer structures. But as far as in the TiO2/Al2O3 bilayer structures electron transport is attributed to the bulk properties of metal oxide films [more details could be found in the reference 15: Andreeva, N.; Ivanov, A.; Petrov, A. Multilevel resistive switching in TiO2/Al2O3 bilayers at low temperature. AIP Advances, 2018, 8, 025208], the top electrode material substitution does not change the band diagram of our bilayers. The introduced oxygen-related electrochemical reactions suppress the electronic mechanism of resistive switching, but does not change the electron transport mechanism in the TiO2/Al2O3 bilayers. Thus, the top electrode material substitution influences only the analog tuning of the resistance of Pt/TiO2/Al2O3/Pt structures, which is associated with oxygen vacancies drifting in under a bias voltage. To provide more details on the ionic-related analog tuning of the resistance state of our structures, we add the missing reference [20], with the detailed description of the oxygen vacancy related formation of conductive filament in Al2O3 layer of the TiO2/Al2O3 bilayer structure (Figure 1 in reference [20]).

  • What does each color represent in figure 1? A proper legend should be added in this figure

We use different colors in the former Figure 1 (new Figure 2) to reflect the difference in the resistance state of the TiO2/Al2O3 bilayer structures at analog tuning. The corresponding color range representation is now supplemented with values of the resistance for different resistance states.

  • How many switching cycles are presented in figure 3(b)?

Five consequent switching cycles are presented in Figure 3 (b), what is now reflected in the figure caption.

  • Authors should add some numbers from the obtained data in the conclusion section instead of only writing general statements

The Conclusions section was rearranged in accordance with the referee’s comment. Numbers for obtained tuning ranges for metal-oxide structures were added.

We the authors deeply appreciate again the times and efforts the reviewer generously shared for reviewing our manuscript.

Reviewer 2 Report

Please present in a table performance of the novel memristor realizations in comparison with some reported realizations which are presented in the cited references. This comparison should have next parameters/features such as frequency range, nonvolatile, hot switching, switching energy, switching speed, endurance, suitable for CMOS technology, programming maximum voltage for ON state, programming minimum voltage for OFF state, ON/OFF conductance ratio, isolation for OFF state, insertion loss for ON state, an equivalent model for ON/OFF state, maximum dissipation…

Author Response

Dear Referee,

We highly appreciate your comments and time you spent for reviewing our manuscript. We have taken into account all the critical comments, replied to them, and revised the manuscript accordingly.

Below, we give a detailed, point-by-point reply to all comments. Corresponding changes made in the revised draft of the manuscript are highlighted in red.

Reply to the Second Referee

Please present in a table performance of the novel memristor realizations in comparison with some reported realizations which are presented in the cited references. This comparison should have next parameters/features such as frequency range, nonvolatile, hot switching, switching energy, switching speed, endurance, suitable for CMOS technology, programming maximum voltage for ON state, programming minimum voltage for OFF state, ON/OFF conductance ratio, isolation for OFF state, insertion loss for ON state, an equivalent model for ON/OFF state, maximum dissipation…

Now there are just a few memristive structures with multilevel resistive switching (see, for example, In-memory computing based machine learning accelerators: Opportunities and challenges by Dr. Kaushik Roy, Purdue University, http://www.nsf-pim.com/slides/1-Slides-Kaushik%20Roy.pdf) and it is considered, that passive crossbar arrays with multilevel memristors with more than 4-bits/cell are not reliable yet. In practice, multilevel resistance states are usually obtained in active crossbar arrays (1T1ReRAM).

Also, it should be noted, that for multilevel ReRAM cells not only their realization and assessment poses significant challenges, but also their characterization due to the absence of proper characterization routine [S. Stathopoulos, A. Khiat, A. Serb and T. Prodromakis, "Benchmarking Analogue Performance of Emerging Random Access Memory Technologies," 2018 IEEE International Symposium on Circuits and Systems (ISCAS), 2018, pp. 1-4, doi: 10.1109/ISCAS.2018.8351793].

The next parameters from list suggested by the reviewer: programming maximum voltage for ON state, programming minimum voltage for OFF state, isolation for OFF state, insertion loss for ON state, maximum dissipation, hot switching and frequency range – could be estimated only in the passive crossbar implementation. The aim of our study is to understand the influence of the material of top electrode on the linearity of the multi-level memristive structures based on TiO2/Al2O3 bilayers, what does not require the crossbar integration.

All other parameters, except an equivalent model for ON/OFF state, which are available for the comparison in case of multilevel devices, are presented in the Table 1, included in the main text. An equivalent model for ON/OFF state in case of multilevel memristive systems is an issue for a separate study.

We the authors deeply appreciate again the times and efforts the reviewer generously shared for reviewing our manuscript.

Round 2

Reviewer 1 Report

The authors have satisfactorily answered my questions and followed the recommendations. This article is in a better and publishable form now.